# Effects of Psychostimulants and Antipsychotics on Serum Lipids in an Animal Model for Schizophrenia

**DOI:** 10.3390/biomedicines9030235

**Published:** 2021-02-26

**Authors:** Banny Silva Barbosa Correia, João Victor Nani, Raniery Waladares Ricardo, Danijela Stanisic, Tássia Brena Barroso Carneiro Costa, Mirian A. F. Hayashi, Ljubica Tasic

**Affiliations:** 1Instituto de Química, Universidade Estadual de Campinas (UNICAMP), Campinas 13083-970, Brazil; banny.barbosa@gmail.com (B.S.B.C.); raniery014@gmail.com (R.W.R.); danijela@unicamp.br (D.S.); tassiabrena@gmail.com (T.B.B.C.C.); 2Departamento de Farmacologia, Escola Paulista de Medicina (EPM), Universidade Federal de São Paulo (UNIFESP), São Paulo 04044-020, Brazil; joaoonanii@gmail.com; 3National Institute for Translational Medicine (INCT-TM, CNPq), Faculdade de Medicina de Ribeirão Preto da Universidade de São Paulo (FMRP-USP), São Paulo 14049-900, Brazil

**Keywords:** lipidomics, schizophrenia, animal models, antipsychotics

## Abstract

Schizophrenia (SCZ) treatment is essentially limited to the use of typical or atypical antipsychotic drugs, which suppress the main symptoms of this mental disorder. Metabolic syndrome is often reported in patients with SCZ under long-term drug treatment, but little is known about the alteration of lipid metabolism induced by antipsychotic use. In this study, we evaluated the blood serum lipids of a validated animal model for SCZ (Spontaneously Hypertensive Rat, SHR), and a normal control rat strain (Normotensive Wistar Rat, NWR), after long-term treatment (30 days) with typical haloperidol (HAL) or atypical clozapine (CLZ) antipsychotics. Moreover, psychostimulants, amphetamine (AMPH) or lisdexamfetamine (LSDX), were administered to NWR animals aiming to mimic the human first episode of psychosis, and the effects on serum lipids were also evaluated. Discrepancies in lipids between SHR and NWR animals, which included increased total lipids and decreased phospholipids in SHR compared with NWR, were similar to the differences previously reported for SCZ patients relative to healthy controls. Administration of psychostimulants in NWR decreased omega-3, which was also decreased in the first episode of psychosis of SCZ. Moreover, choline glycerophospholipids allowed us to distinguish the effects of CLZ in SHR. Thus, changes in the lipid metabolism in SHR seem to be reversed by the long-term treatment with the atypical antipsychotic CLZ, which was under the same condition described to reverse the SCZ-like endophenotypes of this validated animal model for SCZ. These data open new insights for understanding the potential influence of the treatment with typical or atypical antipsychotics on circulating lipids. This may represent an outcome effect from metabolic pathways that regulate lipids synthesis and breakdown, which may be reflecting a cell lipids dysfunction in SCZ.

## 1. Introduction

Schizophrenia (SCZ) is a severe, complex and chronic mental disorder (MD), with a serious impact on patients and their families and caretakers. This highly disabling MD imposes an unemployment rate of about 80%, in addition to an important reduction in the life expectancy of patients, which is estimated to be shortened up by about 20 years compared to the general population without psychiatric dysfunctions [1]. In general, SCZ patients are characterized by positive symptoms such as delusions, hallucinations, psychosis, or negative symptoms, which include impaired motivation, reduction in spontaneous speech and social withdrawal, emotional processing/cognitive deficits, and may possibly include impaired neurocognitive deficits, confused speech or behavior alterations [2,3].

The most accepted theory to explain the neurobiology of SCZ is based on abnormalities in neurotransmission, as for instance, the alterations in dopaminergic, serotoninergic, glutamatergic, among other signaling pathways [1,4]. The main drugs used to treat SCZ symptoms are antipsychotic drugs, which are usually employed to normalize the dysfunctions in neurotransmission. Although antipsychotic drugs can control the main symptoms of SCZ, the disease progression is not stopped by long-term treatments with antipsychotics [5]. Another pathway implicated in SCZ pathophysiology is the kynurenine pathway (KP), which involves the tryptophan metabolism [6]. The KP metabolites modulate neurotransmitters related to cognition [7]. Individuals with MDs presented lower levels of kynurenines, which may be involved in cognitive impairment [8], while conversely, SCZ patients showed increased kynurenine levels [6,8]. Therefore, tryptophan–kynurenine metabolism in psychiatric disease is well established, and disturbance of the tryptophan–kynurenine metabolic pathway might be a promising target to unravel the therapeutic effects of psychoactive drugs. Additionally, the KP is also related to lipid metabolism [6,7].

Several studies have pointed to some abnormalities in cell membranes and brain lipids that compromise the structural integrity and functional properties of neurons in patients with MD [9,10]. Moreover, insufficient uptake, excessive breakdown and/or changes in membrane phospholipids composition are all associated with SCZ and dysfunctional synapses [9,10,11]. The analytical evaluation of the effects of drugs currently employed in clinics is of utmost importance for the progress in the knowledge in the field. For this purpose, optimized animal models have the power to contribute for understanding the pharmacological effects of each class of antipsychotics on animal metabolism and, consequently, to the discovery of new pathways underlying complex diseases such as MDs [12].

The Spontaneously Hypertensive Rat (SHR) strain was recognized as a reliable animal model for studying SCZ due to the depicted SCZ-like behaviors, which were reversed by the treatment with typical and atypical antipsychotics [13,14]. It is worth mentioning that these animal behavior alterations following the treatment with antipsychotics were not associated with the high blood pressure of adult SHR [15], although they were associated with differences in biochemical biomarkers in blood serum and brain from SHR compared with normotensive Wistar rats (NWR) [14,16]. Moreover, these altered levels of biochemical biomarkers in SHR relative to NWR were also observed in SCZ compared with healthy control subjects [17,18,19], reinforcing the validity of this animal model for studying pathophysiological pathways associated with this psychiatric disorder.

The power of analytical approaches for lipid metabolism in neuropsychiatric disorders is increasingly recognized [20,21,22,23], and a better understanding of blood lipids could potentially add important knowledge. Nuclear magnetic resonance (NMR) spectroscopy is a powerful analytical tool that provides relevant information for comparison of different samples, and allows the identification of lipids [24]. In the present study, we tested the alterations in the lipid content by comparing the effects of typical haloperidol (HAL) and atypical clozapine (CLZ) antipsychotic drugs after 30 days of treatment of a validated animal model for studying SCZ (namely SHR), which were compared with a control normal strain (namely NWR). In addition, to mimic the increases of dopamine release, expected to occur in episodes of psychosis, NWR animals were challenged by acute administration of psychostimulants–amphetamine (AMPH) or lisdexamfetamine (LSDX) [16]—for blood lipid contents evaluation. Herein, the lipidomics by NMR analyses aimed to identify potential changes in lipids that could provide insights into the metabolic consequences of the pharmacological interventions with pro-psychotic psychostimulants or antipsychotics employing animal models. These results may contribute to the understanding of the metabolic effects of the treatments of SCZ patients with typical or atypical antipsychotics under clinical conditions.

## 2. Materials and Methods

### 2.1. Animals

Spontaneously Hypertensive Rat (SHR) and normotensive Wistar rat (NWR) strains were treated under previously described conditions [13,14,15,16]. Male 4–5 months-old animals, from our own local colony, were housed in groups of 3–4 animals per cage (41 × 34 × 16.5 cm^3^), under controlled temperature (22–23 °C) and 12/12 h light/dark cycle conditions, with lights on at 07:00 AM, and with free access to water and a normocaloric Nuvilab CR-1 irradiated diet (Quimtia^®^, Curitiba, Brazil). The animals were maintained following the guidelines of the Committee on Care and Use of Laboratory Animal Resources, National Research Council, USA. This study was approved by the Ethical Committee of the Universidade Federal de São Paulo (UNIFESP/EPM), identification CEUA Nº 7290170315, approved on 15 March 2015.

### 2.2. Reagents and Drugs

The solvents (chloroform, methanol, and acetone) used for lipids extraction were from LabSynth Products Laboratories (Diadema, SP, Brazil), and deuterated chloroform (CDCl_3_, with 99.8% of D) was from Cambridge Isotope Laboratories, Inc. (Tewksbury, MA, USA). Other reagents were of analytical grade from Sigma-Aldrich (St. Louis, MI, USA). Antipsychotics haloperidol (HAL, Sigma-Aldrich, St. Louis, MO, USA) and clozapine (CLZ, Pinazan, Laboratório Cristália, São Paulo, Brazil), as well as the psychostimulants amphetamine (AMPH, Sigma-Aldrich) and lisdexamfetamine dimesylate (LSDX, Vynvase™, Shire LLC, São Paulo, Brazil), were dissolved in saline solution, and they were injected by intraperitoneal (ip) route in a volume of 1 mL/kg of animal body weight. The volume of the vehicle administered for the negative controls was also 1 mL/kg of animal body weight.

### 2.3. Drug-Naïve Animals

The blood samples of drug-naïve NWR and SHR male animals (5 months-old) were collected in dry blood tubes, soon after the animal euthanasia by decapitation, strictly following the standards described in the Guidelines for Ethical Conduct in the Care and Use of Animals.

### 2.4. Treatment with Psychostimulants

Male NWR (5 months-old) were grouped in each cage with 4–5 animals, and a single dose of the propsychotic psychostimulants (0.5 or 5.0 mg/kg), namely AMPH or LSDX, was administered by ip route, aiming to mimic the SCZ-like psychotic episodes [13]. Before (baseline) and 2 h after this single ip injection of either psychostimulants, the blood of the animals was collected in heparin tubes by tail punction.

### 2.5. Treatment with Antipsychotics

Male NWR and SHR (4 months-old) animals were kept in cages for a month to acclimate before starting the daily treatment for 30 days. This treatment was performed exactly as previously described to evaluate and reverse the characteristic SCZ-like behavioral and biochemical changes [13,14,16]. Then, animals were grouped into: Group I—control animals receiving vehicle (saline 0.9%, 0.1 mL/kg, ip); Group II: animals treated with HAL (0.5 mg/kg, ip); and Group III: animals treated daily with CLZ (2.5 mg/kg, ip). At the end of the treatments, one day after the last administration of antipsychotics (experimental groups) or saline vehicle (negative control group), the blood was collected in dry blood tubes, soon after the euthanasia of animals by decapitation.

Sixty-one animal serum samples underwent lipids extraction and subsequent proton NMR (^1^H-NMR) analysis. The serum samples from animals were:

(a) non-treated animal strains, NWR (control group, N = 4), and SHR (SCZ group, N = 4), with a total animal serum sample equal to 8;

(b) NWR animals challenged with psychostimulants (AMPH or LSDX) and controls receiving saline, in which:

(1) NWR receiving saline 0.9% (N = 5); (2) NWR receiving single administration of 0.5 mg/kg (N = 5) or 5.0 mg/kg (N = 5) of AMPH; (3) NWR receiving single administration of 0.5 mg/kg (N = 5) or 5.0 mg/kg (N = 5) of LSDX, with a total animal serum samples equal to 25;

(c) treatment with antipsychotics for 30 days: (1) NWR receiving saline 0.9% (N = 4); (2) SHR receiving saline 0.9% (N = 4); (3) NWR treated with CLZ (N = 5); (4) NWR treated with HAL (N = 5); (5) SHR treated with CLZ (N = 5); (6) SHR treated with HAL (N = 5), with a total animal serum samples equal to 28.

### 2.6. Extraction of Lipids from Serum Samples and NMR Analysis

Animal serum (0.5 mL) was mixed for 1 min, using a vortex, with 2.4 mL of the solvent mixture composed of methanol: chloroform: sodium chloride solution (0.15 mol/L) in a ratio of 1:2:2 (*v*/*v*/*v*). Then, the mixture was centrifuged for 20 min at 2200× *g* at 10 °C, and the chloroform phase that contained serum lipids was carefully separated from the hydro-alcoholic phase. Chloroform was evaporated and obtained samples were weighted and stored at −20 °C until the analysis by NMR [24].

Lipids (10 mg) were dissolved in 600 µL of 99.8% deuterated chloroform (CDCl_3_, Cambridge Isotope Laboratories, Inc.) and were transferred into the NMR tubes (5 mm) and kept at 4 °C, to avoid the chloroform evaporation and/or lipid oxidation. ^1^H-NMR analyses were conducted in a Bruker Avance III NMR 600 MHz spectrometer equipped with the Triple Resonance Broad Band NMR probe (Bruker Corp., Billerica, MA, USA). ^1^H-NMR spectra were recorded at 25 °C with the acquisition time of 2.66 s, spectral window width of 26.564 Hz, relaxation time decay (relaxation delay) of 2 s, and 128 number of scans.

For quantitative analysis, some representative samples were chosen and prepared by adding 100 µL of the standard solution of 1,2,4,5-tetrachloro-3-nitrobenzene (5 mg/mL, 99.86% purity; Sigma-Aldrich) into a solution of 10 mg of lipids previously dissolved in 500 µL of deuterated chloroform (CDCl_3_) with tetramethylsilane (TMS) [25]. ^1^H-NMR spectra were recorded using 90° pulse sequence at 25 °C, acquisition time of 8.19 s, spectral window width of 9.9955 Hz, 64 k, relaxation delay of 40 s (5 times T_1_), and 56 scans. ^1^H-NMR data were assigned in accordance with the previously reported NMR data for lipids [26].

### 2.7. Data Processing

^1^H-NMR (600 MHz) spectra for statistical and quantitative analyses were first processed using the TopSpin software (Bruker Corp.). Free induction decays were multiplied by a 0.3 Hz exponential multiplication function prior to Fourier transformation; the tetramethylsilane (TMS) signal was calibrated at δ 0.00, and only a zero-order phase correction was allowed.

For statistical analysis of spectra, the binning of 0.04 ppm was applied to spectral data using MestreNova software, and spectra were transformed into a data matrix. The MetaboAnalyst 3.0 platform (http://www.metaboanalyst.ca/faces/home.xhtml accessed on 1 December 2020) was used for principal component analysis (PCA) and partial least squares discriminant analysis (PLS-DA). No data filtering, no sample normalization, and Pareto scaling (mean-centered and divided by the square root of the standard deviation of each variable) were used in data preprocessing. Leave-one-out cross-validation (LOOCV) was applied in PLS-DA. The accuracy, variable importance in projection (VIP) and clustering results shown as heatmaps (distance measure using euclidean, clustering algorithm using ward.D, view options only group aver-ages of top 15 PLS-DA VIP) were also assessed.

For quantitative purposes, specific ^1^H-NMR signals were manually integrated, and the concentrations of omega-3 (L–linolenic acid) and omega-6 (Ln–linoleic acid) type fatty acids were calculated following the method previously reported by others [26,27]. The concentrations of fatty acids were expressed in molar percentages according to Equations (1) and (2).
L*n*% = 100 *×* A_omega 3_/3 *×* A_G_(1)
*L*% = 100 *×* 2 *×* A_omega 6_/3 *×* A_G_(2)
in which A_omega-3_ and A_omega-6_ are the areas of the bis-allylic proton peaks for omega-3 and omega-6 fatty acids, respectively, and A_G_ is the area of the proton peaks of glyceryl groups; L refers to omega-3, linolenic acid, and Ln refers to omega-6, linoleic acid.

For statistical analysis of the ratio of omega-3/omega-6, data analyses were performed using the GraphPad Prism version 7.0 for Windows (GraphPad Software Corp., La Jolla, CA, USA). Standard parametric (Student’s t-test and one-way Analysis of variance, ANOVA) tests were applied accordingly to variables type and distribution, with post-hoc test Dunnett’s for multiple comparisons. All distribution was checked using a Shapiro–Wilk test. All results are expressed as the value of mean ± standard deviation (SD). The significance threshold was considered at *p* ≤ 0.05.

## 3. Results

### 3.1. Identification of Lipids

^1^H-NMR data of serum lipids (Table 1) were assigned according to the peak numbers (1–25), as indicated in Appendix A. Chemical shifts, peak multiplicity, and coupling constants for the 1–25 compounds were checked against databases and lipids NMR libraries. Lipids from the animal serum samples showed peaks of cholesterol, saturated fatty acids, unsaturated fatty acids, i.e., omega-3 and omega-6 fatty acids, phosphocholines, cardiolipins, and sphingomyelines. Additionally, glycerol esters, glycerolipids (triacylglycerols), glycerophospholipids, and saccharolipids were identified in lipid samples of drug-naïve and treated animals, receiving acute administration of psychostimulants or treated for 30 days with typical or atypical antipsychotics. It is worth mentioning that the lipids identified in the present study are in agreement with the previously described down-regulation of phosphatidylcholine [10,28,29], and upregulation of triacylglycerols [30,31] in SCZ patients compared with healthy control volunteers. In addition, low levels of polyunsaturated fatty acids phospholipid content, specifically in phosphatidylcholine and phosphatidylethanolamine, were reported in first-episode psychosis of SCZ [32,33].

### 3.2. Comparison of Lipids among Drug-Naïve NWR and SHR Animals, and NWR Receiving Psychostimulants

Blood serum lipidomes of drug-naïve NWR animals were significantly different compared to drug-naïve SHR or NWR under psychostimulants effects, as presented in Figure 1. In fact, the lipids isolated from drug-naïve NWR and SHR animals were different from each other, and formed distinct groups (Figure 1A, Appendix A). The drug-naïve NWR lipids showed to be richer in unsaturated fatty acids (UFA), cholesterol (chol), phospholipids (PL)-saccharolipids, and choline glycerophospholipids (ChoGpl) (Figure 1C). On the other hand, the serum lipids in SHR showed different patterns with high quantities of PUFA and fatty acids in general (Figure 1C). In addition, the drug-naïve SHR strain presented lower amounts of PL compared with drug-naïve NWR, but with increased amounts of PUFA (Appendix A). Further investigation may clarify if omega-3 PUFA works in compensation of phospholipids loss, as previously described by others [26,27]. The main observation is that the amounts of phospholipids are stable in the SCZ animal model (SHR), but differences in the chemical structures of these phospholipids, as the ratio between omega-3 and omega-6, could be crucial for the disease and treatments, as we will further discuss.

The lipids from drug-naïve NWR animals suffered alterations under the effects of psychoactive drugs, and they were exposed to psychostimulants (AMPH or LSDX) at different doses (0.5 or 5.0 mg/kg), forming the sub-groups NWR-AMPH*, NWR-AMPH**, NWR-LSDX*, and NWR-LSDX** (Figure 1B,D). Cross-validation of the obtained model is presented in Appendix A. Curiously, serum lipids from the AMPH and LSDX were more similar when the dose dependence was analyzed, and lower doses of psychostimulants determined different effects on NWR lipids compared to higher doses (Figure 1D). However, similarly to the findings for the differences between drug-naïve NWR vs. SHR strains, the ^1^H-NMR signals representing the phospholipids (PL) were less intense in NWR animals receiving AMPH or LSDX compared with control drug-naïve NWR. Additionally, NWR under psychostimulant effects presented higher levels of PUFA. The evaluation of omega-3 and omega-6 concentrations allowed calculating the omega-6 to omega-3 ratio (Table 2). Excessive amounts of omega-6 polyunsaturated fatty acids (PUFA), and high omega-6/omega-3 ratio are both often associated with eicosanoids production in many diseases [34].

The animals receiving LSDX presented the highest concentrations of omega-6 among all studied groups, although the omega-6/omega-3 ratio was not different from NWR receiving AMPH. Moreover, NWR animals receiving psychostimulants showed a remarkable different blood serum lipidome patterns compared with untreated drug-naïve NWR, with a dose-independent decrease in omega-3, as observed for 10-fold different doses of psychostimulants (0.5 and 5.0 mg/kg).

It is important to point that the decrease of omega-3 and of omega-6 acids were statistically significant (*p* = 0.0019 and *p* = 0.0002, respectively), as well as the omega-6/omega-3 ratio (*p* = 0.0042), compared with drug-naïve SHR or drug-naïve NWR. However, only omega-3 (decrease) and omega-6/omega-3 (increase) ratio were significantly changed (*p* < 0.0001) in NWR after administration of psychostimulants. Thus, we suggest that the omega-3 acid levels could be used as a parameter to evaluate the effects of psychostimulant drugs on lipid changes in animal models.

### 3.3. Influences of Antipsychotics HAL and CLZ on Lipids in SHR and NWR Animals

The PLS-DA based on the ^1^H-NMR spectra of lipids extracted from the serum of animals treated with typical HAL or atypical CLZ antipsychotics showed significant differences in lipids composition in antipsychotics-treated SHR compared with control SHR receiving vehicle (Figure 2A,C, and Appendix A). The atypical antipsychotic CLZ modified serum lipids to a greater extent relative to the effects determined by the treatment with HAL (Figure 2A), with greater variations in the most chemical shifts (Figure 2C, VIP) compared with the effects determined by the treatment of SHR animals with HAL. Additionally, increases in 4/15, and decreases in 11/15 lipids levels after treatment with CLZ (see dendogram in Figure 2C) were among the most prominent effects. Analysis in the variations in PUFA and omega-3 chemical shifts (Appendix A) showed important decreases with CLZ treatment. It is also important to point to omega-3, and phospholipids (PL) levels decrease in SHR strain, following the treatment with CLZ (Figure 2C), in addition to the decrease in omega-6 (0.84 ppm), and increase in fatty acids (1.24 ppm). However, an increment in membrane omega-6 fatty acids in SCZ patients after treatment with CLZ was described by others [35].

Moreover, CLZ treatment showed the greatest lipids variations for comparisons between antipsychotic-treated and control NWR animals receiving vehicle (Figure 2B,D, Appendix A). Treatment with HAL slightly increased PUFA and phospholipids (PL) in NWR. In addition to the decreases in PUFAs levels (particularly omega-6), and phospholipids levels observed after the treatment with CLZ, choline glycerophospholipids (ChoGpl) were also identified as an important lipid subclass of phospholipids to distinguish the effects of CLZ in NWR, as ChoGpl were also decreased by CLZ treatment (Figure 2D and Appendix A). This last effect was opposite to that observed for SHR strain treated with CLZ, as illustrated in Appendix A, in which ChoGpl increased in SHR animals after CLZ treatment. In addition, the CLZ effect on omega-3 and omega-6 levels and their ratio in NWR strain were also observed. CLZ determined decreases in omega-3 and omega-6 in both NWR and SHR strains.

The treatment with the atypical antipsychotic CLZ modified the blood serum lipid profiles in SHR animals by causing a different trend compared with what was observed in NWR. Changes in omega-3 and omega-6 determined by the treatment with (typical or atypical) antipsychotic drugs in NWR and SHR (mean values ± SD) are shown in Table 3.

The levels of omega-3 and omega-6 were both significantly different between NWR and SHR lipidomes (*p* = 0.0019 and *p* = 0.0002, respectively). The levels of omega-3 decreased with the treatment with antipsychotic drugs (HAL or CLZ) (*p* < 0.0001), while the omega-6/omega-3 ratios were significantly increased in SHR and NWR after the treatment with HAL or CLZ.

The levels of omega-3, omega-6, and phospholipids may suggest that these important PUFAs were incorporated into ChoGpl, which play important roles in cell membranes. ChoGpl were decreased in NWR treated with CLZ, and were increased in SHR treated with CLZ, and ChoGpl levels in SHR after the treatment with this atypical antipsychotic were closer to those in control NWR receiving vehicle (Appendix A).

The differences in the effects of typical and atypical antipsychotics were evidenced by the greater lipid changes observed for the treatment with CLZ compared with HAL, as one could expect based on the well-known general pharmacological superiority of CLZ compared to typical antipsychotics, as reported by many [36,37,38]. In addition, CLZ is approximately 30% more effective in controlling schizophrenic episodes in treatment-resistant patients than other antipsychotic drugs [1]. However, both typical HAL and atypical CLZ antipsychotic drugs equally induced changes in omega-6/omega-3 values in both NWR and SHR strains (Table 3).

## 4. Discussion

The presence of cholesterol in the serum samples from SCZ-animal model studied here could be associated with the accumulation of cholesterol in the nigrostriatal pathway, which was suggested to contribute to the dopaminergic neurodegeneration in mice brain [39], or to induce cognitive dysfunctions in rats [40]. Additionally, dysfunctions in brain cholesterol homeostasis have been extensively correlated to several other brain disorders, such as autism, Alzheimer’s, Parkinson’s and Huntington’s diseases [35,41,42,43].

In addition, polyunsaturated fatty acids (PUFAs) are reported to be significantly correlated with the negative SCZ symptoms, and their incorporation into plasma membrane phospholipids can remodel the molecular organization of cholesterol-enriched lipid microdomains [44,45]. Some changes in the composition of membrane phospholipids could be associated with SCZ, since the abnormal composition of esterified fatty acids, such as phospholipids, has been reported in plasma, red blood cells, fibroblasts and in *post-mortem* prefrontal cerebral cortical tissues of SCZ patients. Then, the storage and release of neurotransmitters may also be affected by the changes in the lipid composition of neuronal cell membranes [9,10,11,28,30,46,47,48,49]. Interestingly, omega-3 PUFA deficiency in blood was reported to be a trigger of the effects on the dopamine system, and this blood PUFA deficiency was also associated with SCZ [50]. In addition, omega-3 PUFA deficiency was associated with cognitive impairment, which directly impacts the social functioning in patients with SCZ [51]. Thus, the decreases of phospholipids in SHR animals, and also in NWR animals after receiving psychostimulants (such as AMPH and LSDX) possibly validate the SCZ animal model adopted here. However, the decreased amounts of omega-3 and omega -6 fatty acids in SCZ animal models compared with control NWR strain could point out to possible compensation mechanisms [52].

The differences between animals receiving AMPH or LSDX, including the highest concentrations of omega-6 observed after LSDX administration, could be explained by the fact that psychostimulant effects of AMPH are immediate, while LSDX is a prodrug of AMPH and requires its conversion to the active metabolite AMPH, explaining the differences in their action onset [53]. However, despite the same mechanisms of action of these psychostimulants, they also differ in the determination of blood pressure increases, which is more evident for AMPH compared with LSDX, even when used at same doses [1], but which was not correlated with the changes in biochemical biomarkers or in SCZ-like animal behavior [13,14,15,54]. Taking this into account, two different doses of psychostimulants were evaluated here, as they could potentially lead to different effects on lipids metabolism, as we, in fact, observed here.

The linoleic (omega-6) and alpha-linolenic (omega-3) fatty acids are two essential PUFAs that must come from diet, and they are also constituents of neuronal membrane phospholipids with a reported contribution to proliferation and differentiation of neural stem cells [55]. Moreover, the importance of omega-3 fatty acids for the structure and function of neuronal membranes is also well-known [56,57]. Therefore, the increased levels of omega-3 in drug-naïve control NWR compared to drug-naïve SHR strain may point out a possible “healthier” cerebral condition in NWR animals. There are some recognized differences in susceptibility to psychostimulant-induced neurotoxicity [58], as for instance, the psychostimulant-induced release of dopamine into the extracellular space, from the newly synthesized pool of transmitter, which plays an essential role in drug-induced neurotoxicity [59]. And this may possibly explain the differences observed for the effects of different doses of AMPH or LSDX evaluated here. Furthermore, lipid metabolism in SCZ might be related not just to the aberration of neural pathways, but also to disturbances in the tryptophan–kynurenine metabolic pathway [6,7,8], which may warrant future investigation.

Again, since phospholipids are essential constituents of the brain cell membranes, their metabolism might be of great importance in SCZ, as the structural integrity and functional properties of neurons are strongly affected in SCZ patients [9,10]. Moreover, insufficient uptake and biosynthesis or even excessive breakdown of phospholipids from the brain membrane were all hypothesized to be associated with SCZ and dysfunctional synapses [9,10,11]. Thus, the greater levels of phospholipids in the SCZ animal model treated with antipsychotics HAL or CLZ might be indicative of a possible de novo stimulated synthesis of phospholipids aiming to supply the deficiency of these phospholipids.

Ward and collaborators suggested that the exposure to atypical antipsychotics may not differentiate metabolic phenotypes of patients with SCZ [36], in spite of the several other reports suggesting the neurotoxicity of antipsychotics [37,38,39,60]. In addition, we also need to consider that these antipsychotics could possibly inhibit the intracellular traffic of lipids [61]. Therefore, in general, we can suggest that typical and atypical antipsychotic drugs had opposite effects on lipid changes, as both HAL and CLZ increased the relative concentrations of omega-6/omega-3 ratio in SHR, although also increasing the levels of ChoGpl.

Considering the well-known importance of omega-3 fatty acids for the structure and function of neuronal membranes [56,57], the decreases in omega-3 and phospholipids in SHR after the treatment with HAL may suggest a possible negative effect of typical antipsychotics in the lipid profile of SCZ patients. However, the increases in omega-6/omega-3 ratios, and the decreases of omega-3 and phospholipids (mainly ChoGpl) levels after the treatment with CLZ, may both suggest a good correlation with the recognized general pharmacological superiority of CLZ compared to typical antipsychotics [36,37,38]. Moreover, more evident changes in lipid profiles in both rat strains were observed for the treatments with the atypical antipsychotic CLZ compared with typical antipsychotic HAL.

## 5. Conclusions

Taken together, the results presented here support a concept of altered lipids metabolism in a validated animal model for schizophrenia (SCZ), namely SHR. Acute administration of psychostimulants in control animal strain (NWR) showed similar lipid composition alterations as observed in SHR serum samples, and as also observed in the first episode of psychosis or SCZ patients. In addition, the SCZ-like serum lipids profile was reversed more efficiently by the long-term treatment with the atypical antipsychotic drug clozapine (CLZ), relative to the typical antipsychotic haloperidol (HAL). Considering that CLZ is approximately 30% more effective in controlling schizophrenic episodes in treatment-resistant patients than other antipsychotic drugs, it would be possible to hypothesize that CLZ effects on SCZ symptoms could also benefit from the serum lipids alterations as described herein. Moreover, the treatment with the HAL did not determine significant turnover in serum lipids in the present animal model for SCZ, as the lipids remained almost unaltered even after long-term treatment with this typical antipsychotic drug. Therefore, although metabolic syndrome has been more often correlated with the long-term treatment with atypical antipsychotic drug CLZ, it seems to be a better option to minimize the alteration in serum lipids in SCZ patients.

## Figures and Tables

**Figure 1 biomedicines-09-00235-f001:**
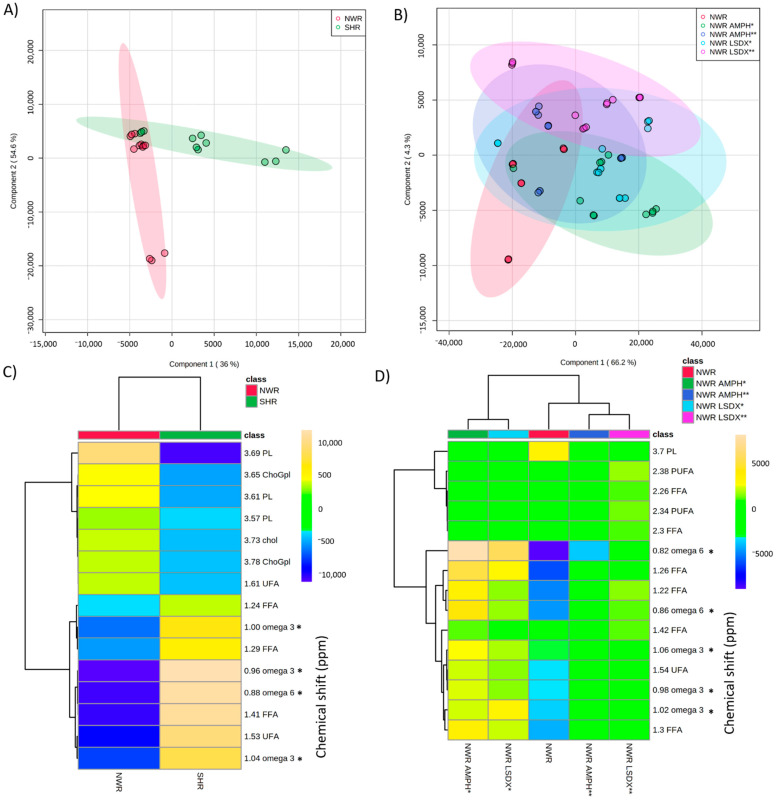
Partial least squares discriminant analysis (PLS-DA) on animal serum lipids ^1^H-NMR data: (**A**) schizophrenia (SCZ) model (spontaneously hypertensive rat, SHR) vs. control (normotensive Wistar rat, NWR) score plot in PC 1 and PC 2. A total of 23 spectra were used for PLS-DA analysis of SHR against NWR, being four samples of each treatment in triplicate excluding one outlier (SHR group). (**B**) 3D score plot for evaluation in NWR of the effects on lipids of psychostimulants amphetamine (AMPH) or lisdexamfetamine (LSDX) with doses of (*) 0.5 mg/kg and (**) 5.0 mg/kg. A total of 62 spectra were used for PLS-DA analysis of NWR after AMPH or LSDX administration, with five samples for each treatment (four samples for NWR), performed in triplicate, but excluding three outliers in NWR-LSDX* group, three outliers in NWR-AMPH* group, two outliers of NWR-LSDX** group, and two outliers in NWR-AMPH** group. (**C**) Heatmap for SHR vs. NWR shows relative concentrations of the 15 most important variables in projection (VIP). (**D**) Heatmap for NWR vs. AMPH or LSDX shows the relative concentrations of the fifteen variables (VIP) before and after psychostimulants administration. In (**C**,**D**), chemical shifts marked for omega 3 *, omega 6 *, may come from other acyl groups. PL: phospholipids, ChoGpl: choline glycerophospholipids, chol: cholesterol, FFA: free fatty acids, PUFA: polyunsaturated fatty acids, UFA: unsaturated fatty acids.

**Figure 2 biomedicines-09-00235-f002:**
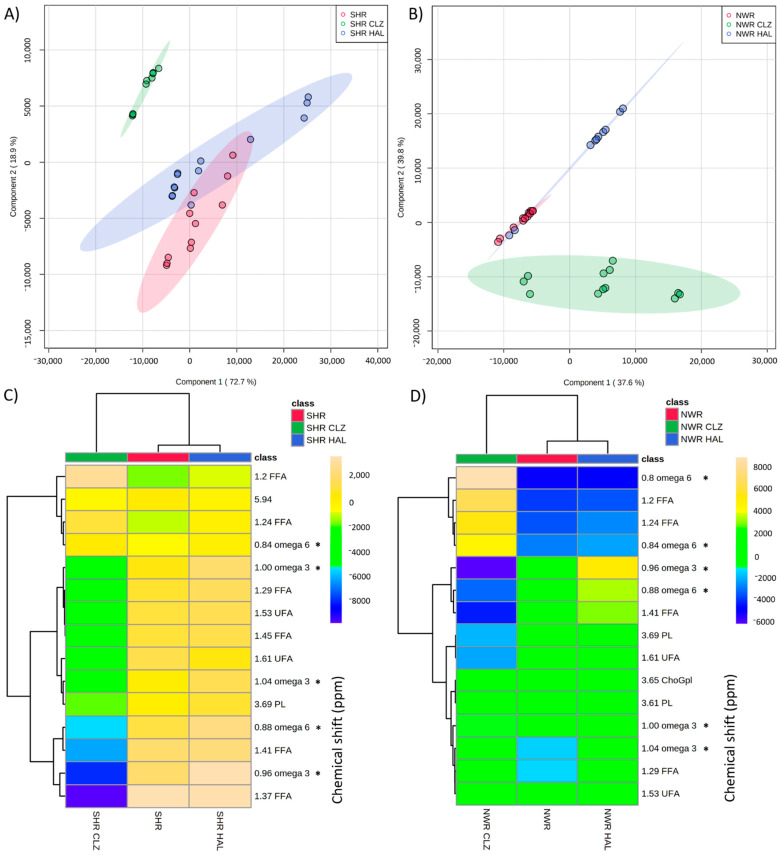
PLS-DA charts obtained for ^1^H-NMR of lipids isolated from drug-naïve SHR (**A**,**C**) and drug-naïve NWR animals (**B**,**D**) after the treatment with typical haloperidol (HAL) or atypical atypical clozapine (CLZ) antipsychotic drugs. SHR lipid profile changes are shown as PLS-DA scores (**A**) or heatmap (**C**) with the relative concentrations of the 15 best-ranked chemical shifts (VIP scores). NWR lipid profile changes are shown as PLS-DA scores (**B**) or heatmap (**D**) with the relative concentrations of the 15 best-ranked chemical shifts (VIP scores). A total of 34 spectra were used for PLS-DA analysis of SHR after treatment with typical HAL or atypical CLZ antipsychotic drugs, with five samples for each treatment (4 samples for SHR), performed in triplicate, but excluding one outlier of SHR, five outliers of SHR CLZ, and two outliers of SHR-HAL. A total of 34 spectra were used for PLS-DA analysis of NWR treated with typical HAL or atypical CLZ antipsychotic drugs, with five samples for each treatment (four samples for NWR), performed in triplicate, but excluding two outliers of NWR-CLZ, and five outliers of NWR-HAL. In (**C**,**D**), chemical shifts marked for omega 3 *, omega 6 *, may come from other unsaturated acyl groups. PL: phospholipids, ChoGpl: choline glycerophospholipids, chol: cholesterol, FFA: free fatty acids, PUFA: polyunsaturated fatty acids, UFA: unsaturated fatty acids.

**Table 1 biomedicines-09-00235-t001:** Rat serum ^1^H-NMR spectral assignments. The NMR peaks were numbered as illustrated in Appendix A. Legend: ^a^ omega-6, ^b^ omega-3, and ^c^ glycerol (see Equations (1) and (2) in Methods Section).

Peak	Chemical Shift (ppm)	Assignment
**1**	0.58–0.70	Terminal methyl group in cholesterol -C**H**_3_
**2**	0.75–1.00	-C**H**_3_ protons of saturated, oleic and linoleic acyls (omega-6)
**3**	0.93–1.02	-C**H**_3_ protons of linolenyl chain (omega-3)
**4**	1.20–1.50	Methylene protons of aliphatic chains -(C**H**_2_)_n_
**5**	1.50–1.75	β–methylene protons of the carbonyl –OC(O)-CH_2_-C**H**_2_-
**6**	1.95–2.10	Methylene protons in the α-position of double bonds –C**H**_2_-CH=CH-
**7**	2.20–2.50	Methylene protons in the carbonyl α-position –OC(O)-C**H**_2_-
**8**	2.70–2.84	C**H**_2_-*bis*-allyllic protons of polyunsaturated fatty acid (PUFA) chains
**9**	2.80–2.90 ^a^	Divinyl methylene protons =HC-C**H**_2_-CH= of omega-6 including linoleyl chain
**10**	2.79 ^b^	Divinyl methylene protons =HC-C**H**_2_-CH= of omega-3 including linolenyl chain
**11**	3.10–3.20	Methylene protons α to the heteroatom –C**H**_2_-OH
**12**	3.20–3.40	Methyl protons of charged nitrogen –^+^N(C**H**_3_)_3_
**13**	3.40–3.60	Heteroatom proton –O**H**
**14**	3.44–3.59	CH of cholesterol relative to the C-3 proton
**15**	3.50–3.85	Methylene protons α to a charged nitrogen C**H**_2_-N^+^(CH_3_)_3_
**16**	3.65–3.75	Hexoses protons on α-carbon to the heteroatom
**17**	3.88	Methine proton at C-4 of galactose
**18**	4.00–4.30	Protons on α-carbon to the heteroatom
**19**	4.10–4.40	Protons on α-carbon to the heteroatom (OH) and β to the amine -O-C**H**_2_-CH_2_-N^+^(CH_3_)_3_
**20**	3.90–4.40	Methylene protons α to the heteroatom in phosphorus C**H**_2_-O-P
**21**	4.10–4.30 ^c^	Sn-1 and Sn-3 protons of glycerol -C**H**_2_-OC(O)R
**22**	5.00	Anomeric carbon protons of galactose
**23**	5.20–5.40	Amine protons –**H**N(CH_3_)_2_
**24**	5.25–5.50	Sn-2 protons of glycerol > C**H**-O-C(O)R
**25**	5.27–5.38	Protons of double bonds with conformation Z-C**H**=HC-

**Table 2 biomedicines-09-00235-t002:** Changes in omega-3 and omega-6 acids determination using the ^1^H-NMR data from control NWR animals after the administration of psychostimulants (mean values ± SD).

		Omega 3 (%)	Omega 6 (%)	Omega 6/3
**Groups**	NWR	50.16 ± 9.04	25.74 ± 5.18	0.52 ± 0.05
	SHR	26.37 ± 5.39	7.43 ± 1.06	0.29 ± 0.10
	NWR-AMPH *	5.41 ± 1.56	22.28 ± 7.68	4.12 ± 1.30
	NWR-AMPH **	6.03 ± 2.52	25.86 ± 3.10	4.66 ± 1.25
	NRW-LSDX *	10.10 ± 3.75	43.50 ± 17.40	4.33 ± 0.81
	NRW-LSDX **	10.13 ± 2.55	33.65 ± 6.05	3.49 ± 1.05
***p*** **-Values**	NWR × SHR	0.0025 (*t* = 5.24)	0.0002 (*t* = 6.85)	0.0042 (*t* = 4.16)
	NWR × NWR-AMPH *	<0.0001 (*t* = 10.90)	0.5400 (*t* = 0.83)	0.0003 (*t* = 6.15)
	NWR × NWR-AMPH **	<0.0001 (*t* = 10.51)	0.9992 (*t* = 0.042)	< 0.0001 (*t* = 7.41)
	NWR × NWR-AMPH * + NWR-AMPH **	<0.0001 (F (2, 14) = 109.0)	0.5403 (F (2, 14) = 0.64)	< 0.0001 (F (2, 14) = 23.26)
	NWR × NWR-LSDX*	<0.0001 (*t* = 9.14)	0.0467 (*t* = 2.57)	< 0.0001 (*t* = 10.42)
	NWR × NWR-LSDX **	<0.0001 (*t* = 9.52)	0.4440 (*t* = 2.31)	< 0.0001 (*t* = 6.28)
	NWR × NWR-LSDX * + NWR-LSDX **	<0.0001 (F (2, 14) = 78.27)	0.0749 (F (2, 14) = 3.24)	< 0.0001 (F (2, 14) = 33.74)
	SHR × NWR-AMPH * + NWR-AMPH **	<0.0001 (F (2, 14) = 54.64)	0.0005 (F (2, 14) = 16.30)	0.0002 (F (2, 14) = 20.37)
	SHR × NWR-LSDX * + NWR-LSDX **	<0.0001 (F (2, 14) = 24.42)	0.0016 (F (2, 14) = 12.18)	< 0.0001 (F (2, 14) = 29.97)

Note: Normal Wistar rat (NWR); Spontaneously Hypertensive rat (SHR); amphetamine (AMPH) and lisdexamfetamine (LSDX) (* 0.5 and ** 5.0 mg/kg); standard deviation (SD). Student’s t-test for NWR × SHR; NWR × NWR-AMPH *; NWR × NWR-AMPH **; NWR × NWR-LSDX * and NWR × NWR-LSDX **. One-way ANOVA, post-hoc test Dunnett’s for multiple comparisons for NWR × NWR-AMPH * × NWR × AMPH **, NWR × NWR-LSDX * × NWR × LSDX **, SHR × NWR-AMPH * × NWR × AMPH ** and SHR × NWR-LSDX * × NWR × LSDX **. Values are significantly different for *p* ≤ 0.05 (N = 5).

**Table 3 biomedicines-09-00235-t003:** Changes in omega-3 and omega-6 determined by ^1^H-NMR in NWR and SHR after the treatment with (typical or atypical) antipsychotic drugs (mean values ± SD).

		Omega-3 (%)	Omega-6 (%)	Omega 6/3
**Groups**	NWR	50.16 ± 9.04	25.74 ± 5.18	0.52 ± 0.05
	SHR	26.37 ± 5.39	7.43 ± 1.06	0.29 ± 0.10
	NWR-HAL	0.75 ± 0.26	2.15 ± 0.54	3.02 ± 0.71
	SHR-HAL	0.92 ± 0.09	2.96 ± 0.77	3.16 ± 0.62
	NWR-CLZ	0.90 ± 0.44	2.69 ± 1.48	2.87 ± 1.42
	SHR-CLZ	1.00 ± 0.57	1.07 ± 0.50	2.60 ± 2.08
***p*** **-Values**	NWR × SHR	0.0019 (*t* = 5.24)	0.0002 (*t* = 7.83)	0.0042 (*t* = 3.74)
	NWR × NWR-HAL	< 0.0001 (*t* = 14.05)	<0.0001 (*t* = 11.48)	0.0037 (*t* = 6.89)
	NWR × NWR-CLZ	< 0.0001 (*t* = 14.03)	<0.0001 (*t* = 10.71)	0.0163 (*t* = 3.31)
	NWR × NWR-HAL + NWR-CLZ	<0.0001 (F (2, 14) =195.4)	<0.0001 (F (2, 14) = 114.4)	0.0708 (F (2, 14) = 3.41)
	SHR × SHR-HAL	<0.0001 (*t* = 10.73)	0.0002 (*t* = 7.27)	0.0110 (*t* = 8.92)
	SHR × SHR-CLZ	<0.0001 (*t* = 10.68)	<0.0001 (*t* = 10.75)	0.0456 (*t* = 2.66)
	SHR × SHR-HAL + SHR-CLZ	<0.0001 (F (2, 14) = 114.6)	<0.0001 (F (2, 14) = 64.81)	0.0143 (F (2, 14) = 6.69)

Note: Normal Wistar rats (NWR); Spontaneously Hypertensive rats (SHR); haloperidol (HAL); clozapine (CLZ); standard deviation (SD). Student’s *t*-test for NWR × SHR; NWR × NWR-HAL; NWR × NWR-CLZ; SHR × SHR-HAL and SHR × SHR-CLZ, and One-way ANOVA, post-hoc test Dunnett’s for multiple comparisons for NWR × NWR-HAL × NWR-CLZ and SHR × SHR-HAL × SHR-CLZ. Values are significantly different for *p* ≤ 0.05 (N = 5).

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
