# Peer review of "Effects of Psychostimulants and Antipsychotics on Serum Lipids in an Animal Model for Schizophrenia"

_biomedicines, 2021, doi:10.3390/biomedicines9030235_

Round 1

Reviewer 1 Report

The study is very interesting, it refers to a rarely undertaken topic concerning the role of cell membrane composition in the pathogenesis of mental illness. The experiment described by the authors was properly planned and described, the results are clearly presented and do not raise my doubts. The references are well suited to the topic of the work.

I do not agree that the main conclusion of the last sentence of Discussion section: “The present study is the first to introduce a translational approach for understanding the influences of the treatment with typical or atypical antipsychotics on the metabolic syndrome, which is often reported in patients with schizophrenia. " In my opinion, this conclusion is secondary, but the main conclusion should be that atypical antipsychotics can exert their antipsychotic effects by normalizing the composition of membrane lipids.

The abbreviations of drug names that are commonly used are different than those introduced by the authors. I suggest replacing them with: HAL (haloperidol) and CLZ (clozapine).

Author Response

Responses for review #1:

Author's Reply to the Review Report (Reviewer 1)

The study is very interesting, it refers to a rarely undertaken topic concerning the role of cell membrane composition in the pathogenesis of mental illness. The experiment described by the authors was properly planned and described, the results are clearly presented and do not raise my doubts. The references are well suited to the topic of the work.

Answer: 

Thank you. We are grateful for the reviewer's comments.

The abbreviations of drug names that are commonly used are different than those introduced by the authors. I suggest replacing them with HAL (haloperidol) and CLZ (clozapine).

Answer:

Thank you. We have replaced the abbreviations and used them throughout the manuscript HAL for the haloperidol, and CLZ  for the clozapine.

In my opinion, this conclusion is secondary, but the main conclusion should be that atypical antipsychotics can exert their antipsychotic effects by normalizing the composition of membrane lipids.

Answer:

Yes, we agree. Thank you. The conclusion section has been modified, and the findings of the study were presented more directly.

Reviewer 2 Report

The study evaluated the serum lipids of a suggested animal model for schizophrenia (spontaneously hypertensive rat strain), and a normal control rat strain, after long-term treatment with typical (haloperidol) or atypical (clozapine) antipsychotics. In addition, psychostimulants, amphetamine and lisdexanfetamine, were administered to animals aiming to mimic the human first episode of psychosis, and the effects on lipids were also evaluated. The study is interesting and the findings are potentially important; however, there are some issues that need to be addressed.

Major:

Although the spontaneously hypertensive rat (SHR) strain has been suggested as an animal model of schizophrenia it is primarily the model for essential (or primary) hypertension, used to study cardiovascular disease. My major concern is whether this model is good for studding changes in the lipid metabolism in schizophrenia, as lipid profile of SHR strain might be primarily related to the hypertension and not schizophrenia.

Materials and methods Treatment with antipsychotics

Why amphetamine and lisdexanfetamine were administered in two doses whereas haloperidol and clozapine were administered in one does? How the authors have chosen these doses?

Materials and methods-Data processing

In equations explain symbols L, Ln, AL i Aln.

Please specify which statistical tests form the GraphPad Prism were used in statistical analysis of the data.

Results:

The authors wrote „Lipids from the SHR serum showed peaks of cholesterol, saturated fatty acids, unsaturated fatty acids, i.e. omega-3 and omega-6 fatty acids, phosphocholines and cardiolipins, and sphingomyelines.“ Does this mean that this lipid peaks were not observed in NWR animals?

The tables should be uniform and in addition to p-values the statistical tests used should be written.

The data presented in the heatmaps in the Figure 1 and 2 seem contradictory to the percentage data in Tables 2 and 3. Please explain more clearly, what they are presenting.

In general, the results are not clear and really hard to follow. Please re-write them more clearly. Perhaps you can introduce some subheadings.

Supplementary Tables and Figures and especially not clear and understandable. Graph presents lipid peaks of „some“ animal? Tables present mean of data of how many animals?  In tables, some data are in italic and bold. Why? Tables presents comparisons of several groups it is not clear which ones?

Discussion: Please discuss specifically the findings of the study and use the obtained data to recommend antipsychotic treatment in schizophrenia patients.

Author Response

Author's Reply to the Review Report (Reviewer 2)

Major:

Although the spontaneously hypertensive rat (SHR) strain has been suggested as an animal model of schizophrenia it is primarily the model for essential (or primary) hypertension, used to study cardiovascular disease. My major concern is whether this model is good for studying changes in the lipid metabolism in schizophrenia, as the lipid profile of the SHR strain might be primarily related to hypertension and not schizophrenia.

Answer:

Thank you for the observations. Although we understand the criticism, it is important to consider that there is no ideal animal model for SCZ. This might be mainly due to the heterogeneous behavioral symptom characteristics of this psychiatric disorder that may be unique to humans. Even if we consider the traditional approach of establishing an animal model based on the three classic constructs (namely face validity; construct validity; predictive validity), as proposed by Willner in 1984, in practice, however, no animal model fully meets all three proposed criteria. We also believe that animal models should not be simply organisms that resemble human dysfunctions, but that they should reproduce processes by which animals and humans enter this state [Nani et al., 2020. In: Animal Models in Psychiatric Disorder Studies]. 

In the last few years, we have tried to understand the potential role of metabolism alterations in mental disorders, by using several approaches including clinical studies, and also by employing different animal models, such as the nematode Caenorhabditis elegans, which shows conserved neurobiological systems relative to humans, and this animal model was successfully employed by us to show altered animal behaviors and metabolic profile in animals knockout for genes related to SCZ, relative to wild type animals [Monte et al., Prog. Neuropsychopharmacol. Biol. Psychiatry 2019]. 

Although the Spontaneously Hypertensive Rats (SHR) strain could also be considered a relatively non-reliable animal model for studying SCZ at first impression, interestingly, this strain presents several endophenotypes potentially useful to investigate SCZ, which were shown to be aggravated by pro-psychotic manipulations, for instance, by the administration of psychostimulants such as amphetamine or lisdexamfetamine. Besides, these SCZ-like endophenotypes, such as (1) hyperlocomotion which could be associated with positive SCZ symptoms (Calzavara et al., Behav. Brain Res. 2011), (2) deficits in social interaction which could be associated with negative symptoms [Levin et al., Prog. Neuropsychopharmacol. Biol. Psychiatry 2011], (3) alterations in contextual fear conditioning which are as a measure of cognitive symptoms (Calzavara et al., Schizophr. Bull. 2009), and (4) deficits in prepulse inhibition of startle (considered as a “gold standard” behavioral endpoint in neurodevelopmental models for SCZ research), which correlates with information processing deficits, were all reversed by the typical and atypical antipsychotic drugs under the same pharmacotherapy regimen employed in the present work (Levin et al., 2011). 

Moreover, a recent work, using the confirmatory factor analysis (CFA) to hypothesize the relationship between these observed variables, and (one or more) underlying traits, showed the existence of a single schizophrenia-like trait (STL) underlying these behavioral measures, whose application offers an excellent face validity with psychometric studies using this specific SHR strain (Peres et al., 2018. Prog. Neuropsychopharmacol. Biol. Psychiatry). 

Also, recent studies published by our group have demonstrated that treatment with these typical and atypical antipsychotics under this same condition in which SCZ-like animal behavioral deficits reversal was described, no change of the blood pressure of SHR animals are observed, while changes in biochemical markers in SHR treated with antipsychotics mimicked the changes observed in clinical studies, evaluating SCZ patients under treatment with antipsychotics (Nani et al., World J Biol Psychiatry 2019; Nani et al., Sci. Rep. 2020). 

Given this set of evidence, we believe that this animal model can be useful to investigate changes in the metabolic profile related to the pathophysiology of SCZ, and even if the lipid profile of SHR could be primarily related to the hypertension of these animals, the lipid profile changes reported here are determined by the treatment with the typical or atypical antipsychotics, under a pharmacotherapeutic regimen known to not change the blood pressure of the SHR animal, as mentioned above (Nani et al., World J Biol Psychiatry 2019; Nani et al., Sci. Rep. 2020). The text was revised and amended to clarify this point.

Materials and methods Treatment with antipsychotics

Why amphetamine and lisdexamfetamine were administered in two doses whereas haloperidol and clozapine were administered in one dose? How the authors have chosen these doses?

Answer:

As mentioned in the text, SHR animals were strictly treated with antipsychotics under the same conditions previously described to revert the altered SCZ-like behavior of these animals (Calzavara et al., 2009; Calzavara et al., 2011; Levin et al., 2011) and to impose biochemical biomarkers changes similarly as observed in SCZ patients compared with healthy controls in clinical studies (Nani et al., World J Biol Psychiatry 2020; Nani et al., Sci. Rep. 2020). Therefore, the doses employed in this work were chosen based on previous studies demonstrating the effectiveness of these drugs to revert animal behaviors, also in other animal models of SCZ (Feldon J., Biol. Psychiatry 1991; Abdul-Monim et al., Behav. Brain Res. 2006; Weiner et al., Neuropsychopharmacology 2003). 

On the other hand, the psychostimulants amphetamine and lisdexamfetamine were administered in two doses because they differ in the onset action due to the need for LSDX conversion into the active metabolite AMPH, and especially in the determination of blood pressure increases, which is more evident for AMPH compared with LSDX, even when used at same doses (Nani et al., World J. Biol. Psychiatry 2019). Besides, it is worth considering that the changes in blood pressure were not correlated with the changes in biochemical biomarkers or with the SCZ-like animal behavior changes (Calzavara et al., Schizophr. Bull 2009; Calzavara et al., Brain Res. 2011; Levin et al., Prog. Neuro-psychopharmacology Biol. Psychiatry 2011; Nani et al., World J. Biol. Psychiatry 2019; Nani et al., Sci. Rep. 2020). Therefore, different effects on lipids metabolism could be observed for different doses of these psychostimulants, as we have in fact observed here and also in previous works (Nani et al., World J. Biol. Psychiatry 2019), two (ten-fold) different doses of psychostimulants were evaluated here. The text was also revised to clarify this important point raised by the reviewer.

Materials and methods-Data processing

In equations explain symbols L, Ln, AL, and Aln.

Answer: 

In equations, L refers to omega-3, linolenic acid, and Ln refers to omega-6, linoleic acid, and this is now explained in the revised text.

Please specify which statistical tests from the GraphPad Prism was used in the statistical analysis of the data.

Answer: 

Standard parametric (Student’s t-test and one-way ANOVA) tests were applied accordingly to variables type and distribution, with posthoc test Dunnett’s for multiple comparisons. This was amended in the text and also added in the footnotes of the tables where this test was applied. 

The authors wrote, „Lipids from the SHR serum showed peaks of cholesterol, saturated fatty acids, unsaturated fatty acids, i.e. omega-3 and omega-6 fatty acids, phosphocholines and cardiolipins, and sphingomyelins.“ Does this mean that these lipid peaks were not observed in NWR animals?

Answers:

Thank you for the observation. We have expressed our findings in the wrong way. All serum samples showed similar lipid types but whose levels were different for each animal strain and/or treatments with drugs. We have corrected this misleading throughout the manuscript.

The tables should be uniform and besides top-values, the statistical tests used should be written.

Answer: 

The tables were revised and the statistical tests used to calculate the p-values were added. Statistical test values were added under the p-values in the tables, and the information on each test is described in the footnotes.

The data presented in the heatmaps in Figure 1 and 2 seem contradictory to the percentage data in Tables 2 and 3. Please explain more clearly, what they are presenting.

Answers:

Thank you for asking about the differences. We have revised the Figures and Tables and explained the obtained results in more detail.

In general, the results are not clear and really hard to follow. Please re-write them more clearly. Perhaps you can introduce some subheadings.

Answers:

We are grateful for the suggestion and for the opportunity to improve our work. The results section was entirely revised to improve the reading.

Supplementary Tables and Figures and especially not clear and understandable. Does the graph present lipid peaks of „some“animals? Tables present the mean of data of how many animals? In tables, some data are in italic and bold. Why? Tables present comparisons of several groups it is not clear which ones?

Answers:

Thank you for pointing to the discrepancies in data illustration and presentation. We have modified legends of all figures and tables, as well as we provide now more precise and correct explanations. Figure S1 is the mean 1H-NMR spectrum of all SHR animals. 

Tables show the statistical data on models obtained for each analyzed case and must be analyzed together with the Figures from the main document (manuscript). The number of spectra used was specified in the PLS-DA graphs. All tables give the statistical data on PLS-DA models used to discriminate the lipids from the animal serum samples and correspond to the Figures given in text - for example - Table S1 brings the data on Accuracy, R2 and Q2 obtained for the PLS-DA of SHR vs NWR; and Table S2 on data NMR vs amphetamine (AMPH) or lisdexamfetamine (LSDX). Data in bold and italic were replaced with a light grey background and they show the best number of LV (Latent Variable) that are applied for the PLS-DA model. The corresponding number of PC is described in the main document.  

Discussion: Please discuss specifically the findings of the study and use the obtained data to recommend antipsychotic treatment in schizophrenia patients.

Answers:

We are grateful for the suggestion and for the opportunity to improve our work. The discussion was revised accordingly. We have discussed the data accordingly to the suggestions and recommendations, and clozapine was a more effective drug at least regarding the more beneficial serum lipids alterations. 

Round 2

Reviewer 1 Report

Thank you for consideration my minor and major remarks. I recommend the present form to be published.

Author Response

Thank you for considering my minor and major remarks. I recommend the present form to be published.

Answer: Thank you for the suggestions, comments, and criticism, which enabled us to greatly improve the final manuscript version.